# Healthcare systems barriers and strategies for pre-exposure prophylaxis utilization amongst young females in Gauteng province: Registered nurse's perspectives

**Doreen Onkarabile Mudau** [ID]*, **Fhumulani Mavis Mulaudzi, Nombulelo Veronica Sepeng, Rafiat Anokwuru**

Department of Nursing, University of Pretoria, Pretoria, South Africa

* doseane@gmail.com

## Abstract

### Background

Healthcare system present several barriers impacting on Pre-Exposure Prophylaxis (PrEP) utilization amongst young females. To overcome these barriers, there are strategies that could be employed to improve PrEP performance. Within healthcare system, registered nurses play critical role on PrEP utilization, however their perceptions on barriers and strategies have not been explored and described in this setting. Therefore, this paper aimed at exploring, describing and contextualizing registered nurse's perceptions on health systems barriers and strategies regarding PrEP utilization amongst young females.

### Methods

Qualitative, explorative, descriptive and contextual method was used to explore and describing registered nurse's perceptions on healthcare systems barriers and strategies regarding PrEP utilization. Purposively sampled participants for focus group discussions and data was analysed using reflexive thematic analysis.

### Results

Barriers and strategies regarding PrEP utilization were reported as themes which emerged with eight sub-themes. Which included; healthcare system structure, healthcare providers, health promotion and medication barriers and strategies. Healthcare system structural barriers included limited PrEP access, healthcare workers related comprised training, competency and staffing while health promotion included poor awareness, inaccurate PrEP information, HIV-ART related stigma. Pill related barriers were side effects, contraindication and monitoring burden crucial to be addressed to enhance PrEP usage. Moreover, strategies such as increased access, PrEP integration to existing HIV, FP services, Department of Education and training of nurses on PrEP would assist to overcome these barriers. Proactively managing side-effects, increased awareness, using media platforms

**Data availability statement:** All relevant data are within the paper and its Supporting Information files.

**Funding:** the author who received awards: DO Grand names & number: National Research Foundation of South Africa (orcid.org/0000-0002-5603-2920) and University of Pretoria Postgraduate Bursary (20814594). URL: https://www.nrf.ac.za/. The funder had no role in study design, data collection and analysis, decision to publish, or preparation of the manuscript.

**Competing interests:** The authors declare that they have no competing interests.

to disseminate information, quality counselling and three-monthly injectable PrEP for the success of desired utilization amongst young females.

## Conclusion

Findings reported healthcare system structure, healthcare providers, health promotion and medication barriers and strategies affecting utilization of PrEP voiced by registered nurses within focus group discussions.

## Introduction

PrEP is an HIV prevention antiretroviral medications prescribed for uninfected individuals who are at a high risk of contracting HIV infection [1]. While the rate at which patients at increased risk of being infected with HIV accept the initiation of a PrEP pill refers to "uptake" and thereafter, the rate at which those who accepted PrEP pill initiation and constantly continued to take it uninterruptedly refers to "retention" [2]. PrEP was approved by United States Food and Drug Administration in 2012. Following that, the guide on prescription and criterion on which patients qualifies for PrEP pill serving as an official document supporting healthcare professionals in implementing PrEP in practice was issued by Centre for Disease Control and Prevention [CDC] in 2014 [3]. Thus, World Health Organization (WHO) made emphasize on providing "tenofovir disoproxil fumarate-based oral PrEP" and incorporating it in to the healthcare system for key populations such as young females as part of HIV prevention strategy in 2015. However, healthcare system can present various barriers that impact the ability of registered nurses to effectively support the utilization of PrEP amongst young females who accounts for 25% of new HIV incidences globally [4]. Highlighting the significance of addressing major barriers impacting negatively into upscaling the utilization of PrEP and strategies thereof.

Despite the approval of PrEP, only few countries in Sub-Saharan Africa which is the region with the highest prevalence of HIV, have laws or systems in place to implement PrEP [5]. Thus the national experts along with national regulator, the South African Health Products Regulatory Authority (SAHPRA), recommended that healthcare systems and healthcare providers improvement strategies should be implemented to upscale PrEP as they would address challenges or barriers [1,6].

Although SAHPRA emphasized the need to address healthcare systems and the healthcare providers related barriers to improve utilization of PrEP, several studies undertaken globally among male patients, men who have sex with men, gay, cisgender, transgender, and Black cisgender women and men that have emphasized factors influencing the use of PrEP as the utilization remains low. These components included risky sexual activity, partner's HIV status, structural variables (cost, help with doctor visits and medication payments), HIV stigma and relationship status, partner's HIV status, and clinical aspects (side effects). Aside from racial and ethnic inequalities, other obstacles included the PrEP regimen, challenging access to PrEP prescribers, challenges scheduling visits, picking up medications, and communicating with PrEP prescribers. Concerns about the technical quality and sufficiency of PrEP services, lack of awareness and diffusion of PrEP, stigma associated with sexuality, and cultural and social norms that discourage conversation about sexuality [7,8]. The structural ones were far from health care facilities, scheduling conflicts, high drug costs, inadequate health insurance coverage, lack of knowledge of PrEP among medical providers, low perceived risk of HIV, and worry about side effects are some of the issues that patients face. medical engagement is hindered by organizational, logistical, linguistic, and cultural constraints as well as a shortage of competent transgender treatment [9–20]. The meaning drawn from these studies is that there are several factors

affecting PrEP utilization amongst different key populations however did not address healthcare system barriers and improvement strategies from registered nurses' perspective.

Additionally, research carried out in Southern Africa among transgender women and men who have sex with men and individuals in HIV sero-discordant partnerships identified hurdles to the use of the PrEP tablet. These obstacles included a daily dosing schedule that was inconvenient, the need to stop taking 90-day medications, unfavourable reactions from partners, and stigmatizing health services. Individual factors that affect the use of PrEP include awareness of the medicine and belief in its efficacy, fear of HIV, and stigma around the use of the drug [16,17]. Moreover, there have been few studies investigating AGYW's experiences using PrEP, including compliance, persistence, and discontinuation in SA. This study's findings mentioned difficulties taking pills, social opposition, or stressful or unexpected events. Feelings of disappointment or failure since you cannot continue use of PrEP, stigma associated with it, a decline in motivation, cultural challenges such conflicts with coming-of-age traditions, and a lack of family support. These studies confirms that PrEP utilization is affected by multiple elements, however the healthcare system has not been explored to depth from nurses' perspective.

Further studies conducted in SA reported that the country has approximately one in ten people on PrEP medication in 2020 [1] suggesting that utilization of PrEP is remaining low and there is a need to explore the healthcare system barriers and strategies of PrEP utilization. Specific to Gauteng province [21] study reported that, only 7% of Adolescent Girls and Young Women (AGYW) between the age of 15 to 24 years and tested HIV negative were initiated on PrEP over four years period (2017 to 2020). While retention of AGYW already enrolled into PrEP drastically dropped to 27% on month however, there were no recorded barriers and strategies to improve the reported poor PrEP utilization. Suggesting that, performance of both the enrolment and retention amongst AGYW on PrEP pill is lacking behind in this setting, however, did not report any studies addressing barriers and strategies. Hence there is high number of new HIV infection rate amongst key population including young females aged 15 to 24 years. Over and above that, study by [21] highlighted that there is an urgent need of interventions addressing healthcare systems barriers and strategies for PrEP. These interventions should be obtained from registered nurses' perspective to improve PrEP utilization since there are limited studies addressing this gap. Considering that, the "voices" of healthcare professionals particularly registered nurses are golden to explore healthcare systems' barriers and strategies for improved PrEP utilization.

It is in this context considering the above-mentioned aspects, the present study would focus on exploring and describing healthcare system barriers affecting utilization of PrEP amongst young females from registered nurses' perspective. Wherein the focus of the study was in Gauteng Province.

## Materials and methods

This study was approved by the University of Pretoria ethics committee with reference number 244/2021. Moreover, written letter of approval was obtained from Johannesburg health district research committee with reference number GP_202107_007. Clinics' managers and registered nurses voluntarily signed consent forms to participate in this study. Implying that participants in this study consent for their participation through formal written, voluntarily signed and witnessed consent form prior data collection as there were no minors included.

This study employed qualitative, exploratory, descriptive and contextual methods to explore and describe healthcare system barriers affecting utilization of PrEP amongst young females from registered nurses' perspective in Gauteng province. The nature of focus group discussions conducted was explorative, as researchers had no evidence-based knowledge about the registered nurses' perception regarding healthcare system barriers and strategies for PrEP utilization`.

Thus, exploring the participants' lived experiences was important and a descriptive approach is used to observe, describe, and document registered nurses' perceptions as they naturally share them. Non-probability purposive sampling approach was used to purposely select registered nurses with minimum of one year expirience in PrEP counting backwards according to their rotation roster. Meaning the selection of four [3] to eight [22] registered nurses were from the day of data collection counting backwards until the last count. The researchers acknowledges that HIV, including its prevention interventions such as PrEP, are continuously under research and as such, their guidelines change rapidly. Implying that for those who have been offering PrEP services for a year and above, their knowledge was perceived to be sufficient to share quality data. For participants to be recruited, they were also required to be affiliated with statutory body, the South African Nursing Council, working at selected Gauteng Province clinics and voluntarily signed consent form. Prior to data collection, researchers developed focus group discussion guide using available literature. The tool was tested through pilot study (4 registered nurses in two groups) to determine the time needed for each group discussion and establish if the probing questions will report what the study has intended to achieve. Duration of 45 to 60 minutes of discussion using the guide was established as sufficient duration to hold FGD and results of pilot study was included on final findings of this study.

Five focus group discussions were conducted with four to eight registered nurses in each group discusion and total number of participants from the five-focus group discussion was thirty [23] considering five [5] clinics. Data was collected until saturation was reached, which is the point at which the participants do not share more data or no new information is arising when data is collected. All the focus group discussions conducted were audio recorded, field notes were also taken during discussion to complement and reflect the actual perceptions shared. The audio recorder is kept at the research unit within University of Pretoria under strict security and will be kept for 15 years ([until 2038) for verification and audits purposes. These focus group discussions were conducted from the 1ST to 28th of February 2022 and the final analysis was accessible on the 1st of December 2022 for research publication purposes. For confidentiality purpose, participants were attributed by numerical values, implying that they were referred to Participant 01 to participant 30 from the first registered nurse in focus group discussion one up to five. Moreover, the focus group discussions conducted lasted for 60 minutes at average exploring healthcare system barriers and strategies for PrEP utilization amongst young females from registered nurses' perspective. The qualitative data analysis started simultaneously with the data collection. Following that, the recorded focus groups discussions were transcribed and analyzed using reflexive thematic analysis by three data co-coders who independently analysed this data. The co-coders shared findings amongst themselves to reach consensus on the final themes and sub-themes of this study.

Data was collected at clinics that are geographically located within the townships of Gauteng province rendering primary health care services to the economically stable communities or townships surrounded by informal settlements which form part of the communities served in these clinics. However, Clinic E renders healthcare services to only one township because of its geographical location which is very close to the biggest city in the province and is more urban. The care system is accessible covering all age groups, gender, and conditions. These clinics are receiving clinics for all the referring low-volume facilities around them. The clinic with the highest headcount according to the Johannesburg Health District Web-DHIS reported on the 31st of November 2021 was Clinic D [n = 22 545], followed by Clinic B (n = 13 569), then Clinic C (n = 12 666), followed by Clinic A (n = 12 011) and the lowest was Clinic E (n = 11 245).

The researchers employed strategies of trustworthiness to ensure accuracy of the findings through prolonged engagement with participants, wherein two meetings were held before data collection to establish good relationships and explain the purpose of the study in-depth.

Additional to that, findings were shared with participants to confirm all the themes, sub-theme and categories. Moreover, a dense description of research designs and methods were carried out for one to rely on the research findings and study co-coder conducted an audit trail of verbatim scripts, themes, sub-theme and categories derived from data analysis by the researchers. A full description of research methodology, participants' backgrounds and context was done to make it possible for other researchers interested in transferring the study to another context.

## Results

A total of 30 registered nurses participated on five focus group discussions conducted with median age of 40 years (range 31-50), while majority of them being females (n = 25; 83%). Additional to that, majority of them were holding registered nurse (RN) job title (n = 18; 60%), followed by those holding senior registered nurses (n = 10; 33%) and the lowest were those holding clinic manager job title (n = 2; 7%). The majority of these participants had twelve [24] months experience in PrEP (n = 19; 63%) while the lowest were those with thirteen [25] months or more experience in PrEP (n = 11; 37%).

From data collected and analysed, two themes emerged, which included perceived healthcare system barriers of PrEP utilization and strategies to enhance HIV PrEP utilization amongst young females. From these themes, eight [22] sub-themes were determined as outlined in Table 1.

## Theme 1: Healthcare system perceived barriers affecting utilization of PrEP amongst young females

This study explored and described healthcare system barriers affecting utilization of PrEP amongst young females as perceived by registered nurses in five focus group discussions with 30 participants. These barriers were categorized into the following sub-theme, healthcare structure, healthcare provider, health promotion and PrEP medication related barriers. Each of these sub-themes had categories presented in detail below under each sub-theme below. The verbatim transcribed focus group discussion for registered nurses has been quoted under each healthcare system barriers below as outlined in the S1 File. The data was collected using the S2 File. Focus group discussion guide for professional nurses during focus group discussions held.

### Healthcare system structure barriers

This sub-theme emerged from focus group discussions for registered nurses who alluded that, there are healthcare system barriers contributing to low PrEP utilization amongst young females. This sub-theme emerged with category of barrier to access PrEP. This category is outlined in detail below:

Table 1. Registered nurses' perceptions on HIV PrEP utilization amongst young females in Gauteng province.

| Themes | Sub-themes |
|---|---|
| **Healthcare system barriers affecting PrEP utilization** | Healthcare structural barriers |
| | Healthcare provider barriers |
| | Health promotion barriers |
| | PrEP medication barriers |
| **Strategies to enhance utilization** | Healthcare structure strategies |
| | Healthcare provider strategies |
| | Health promotion strategies |
| | PrEP medication strategies |

### Healthcare system structure barrier: Barriers to access PrEP

Participants perceived that there are healthcare system barriers which contribute to low PrEP utilization. The mentioned barriers included barrier to access PrEP as verbalised by Participant 2:

> "…PrEP is not available in all clinics, you see how government implemented PrEP initially, it was wrong, because even if some clinics are expected to give PrEP now, they don't because it's fairly new to them and only clinics which started with it are having nurses who can give and that means, it's not really accessible to all, because now patients must use 6 taxis to come here…" [FG: 3, P2, Female, Age: 33]

Implicit in the above statement is that, the pilot roll-out of PrEP was not an ideal way of implementing it, as it caused other healthcare establishment or clinics not to take responsibility of offering PrEP when they are allowed to do so. It further, created gap knowledge between registered nurses in clinics not offering PrEP at the start of its implementation as it is fairly new building up lack of confidence in them. While the other participant, mentioned that patients are referred from one clinic to another mainly because the clinics which implemented PrEP on the upscaling and not initial stage, would either not order PrEP pill or not have a PrEP competent nurse on duty to offer it to patients thus referring patients from one clinic to another. On this point, participant 4 mentioned:

> "… PrEP accessibility in general is a problem because you can go to clinic A, and there is no PrEP or nurse offering PrEP then they will transfer you or refer you to another clinic and you become discouraged." [FG: 2, P4, Female, Age: 38]

Over and above the above quoted participants, the 24 registered nurses indicated that limited access to PrEP is a crucial healthcare system structure barrier to increased PrEP usage amongst those at risk of HIV including young women. Originated from the stage-by-stage roll-out of PrEP. Thus, most clinics are not offering PrEP despite them mandated to do so.

### Healthcare provider related barriers

Participants indicated that, there are healthcare provider barriers significantly contributing to below the target PrEP utilization rate. This sub-theme emerged with four categories which included: PrEP training, competency barrier, ethical dilemmas, ineffective workflow, shortage of nurses and follow-up and monitoring burden. These categories are outlined in detail below:

### Healthcare provider related barrier: PrEP training and competency barrier

In terms of PrEP training and competency barrier, participants indicated that in most of the clinics expected to offer PrEP, registered in there are either not trained or incompetent on PrEP as a program. This was mentioned to be caused by lack of mentoring or support following the training and or no training at all. Thus, many clinics would have one or two nurses competent enough to offer PrEP. Participant 10 mentioned the following in this regard:

> "…we having four registered nurses trained in this clinic out of 25 registered nurses, do we have enough registered nurses available to assists young people on PrEP in this clinic? No, we don't" [FG: 2, P10, Female, Age: 41]

Other participants verbalised their lack of knowledge of PrEP as nurses and failure to share detailed PrEP information to patients should they require further knowledge clarification.

Hence, nurses often refer patients from either one consulting room to the other or one clinic to the other in cases where not even one nurse is trained or competent to offer PrEP on that specific day. In this regard, participant 29 stated that:

*"I'm not familiar with PrEP especially detailed training, I've heard about it from in-service trainings, but it was just superficial information so I can't say I was trained confidently because there is a lot, I believe I don't know on PrEP, we just give what other nurses told us to give but with no insight, so when patient ask me anything about PrEP, I just refer them to those with knowledge…"* [FG: 5, P29, Female, Age: 39]

Other participants mentioned that, limited training and being less competent affect their willingness to promote PrEP as they would lack detailed knowledge to educate patients thus referring them.

## Healthcare provider related barrier: Ethical dilemmas

Ethical dilemma was mentioned as one of healthcare providers' barriers to good utilization of PrEP amongst young females as indicated by participants 28:

*"What is discouraging is, it now goes into the values, morals and social beliefs as human beings generally that 15-year-old is a child, when you are giving this child this medication that talks more into sexual intercourse, what are you saying to the future of this child? As much as you are protecting the child from possible illnesses that may also hinder the same future, are we not indirectly encouraging this child to continue engaging into unprotected sex, it is a bit of a dilemma. Morally and according to the values, it may not be correct or it may not feel correct. That is what discourages us often…"* [FG: 5, P28, Female, Age: 40]

Suggesting that, the availability of PrEP, nurses being responsible to promote and offer it to a girl child of 15 years may indirectly send message of encouraging the child to practice unsafe sex. Furthermore, other participants shared that, offering PrEP pill to a 15-year-old whom is still a child to them is the same as encouraging their granddaughter to go and have multiple sexual partners. Participant 27 concurred and mentioned that bey saying:

*"Rightfully speaking, a 15-year-old is still a child, whom we still have to remind them to do school homework and now, how do I say take PrEP, to my granddaughter? even if I know that they might be sexually active but really it put us in a tight corner, it's like I am giving her permission to go and sleep around it's a dilemma in a way"* [FG: 5, P27, Female, Age: 30]

## Healthcare provider related barrier: Ineffective workflow and shortage of nurses

Participants shared that, ineffective workflow and shortage of nurses are barriers to increased usage of PrEP amongst young females as verbalised by different registered nurses during focus group discussion:

*"…we do not have enough healthcare workers, nurses mainly and we have so many programs that are supposed to be offered on daily basis. So, when young people come to the clinic for PrEP, they will be sent from pillar to post, or from one consulting room to the other, not because we do not want to give them PrEP but we are short staffed, and only attending patients that are acutely and chronically ill only and obviously they will leave without PrEP…"* [FG: 4, P21, Female, Age: 44]

Implying that, insufficient nursing staff affect utilization of PrEP users mainly because nurses prioritize patients who came for acute and chronic consultation and not prevention program such as PrEP. Often patients wait until the leave the clinic without a PrEP pill. Moreover, concerns around department of health's decision to introduce additional program such as PrEP without adding nurses strongly came out from discussions. Often the imbalance between workload and number of nurses leads to increased waiting time of patient as the other participant mentioned:

*"…we are short staffed, adding PrEP to all other programs we are doing and not adding nurses was a big mistake from government, and not only PrEP but all other additional programs, shortage of nurses makes young females to wait for longer and they will leave, you know them they are impatient." [FG: 3, P17: Male, Age: 39]*

## Health promotion barriers

Participants alluded that, there are health promotion barriers which contribute to low PrEP utilization rate amongst young females. This sub-theme emerged with three categories which included: poor PrEP awareness, inaccurate information dissemination regarding PrEP and HIV-ART related stigma. These categories are outlined in detail below:

### Health promotion barrier: Poor PrEP awareness

Participants alluded that because of poor PrEP awareness, many young females do not visit the clinics to request PrEP emphasizing that other prevention intervention are well known such as condoms however PrEP awareness is still very low in populations in need of it like young females. One participant stated that:

*"I think it's lack of information and knowledge about PrEP. I think we're all aware about preventions, like condoms, but we're not aware of PrEP generally as people. A lot of young people don't know anything about it, what it does and where you can they get it…so awareness is a problem" [FG: 5, P24, Male, Age: 37]*

Indicating that, health promotion is still lacking to address poor awareness of PrEP and thus low number of young females using PrEP. Other participants shared that, PrEP pill is quite new in many settings and therefore, awareness is lacking and stated that:

*"You know PrEP is fairly new, not only to young females but to everyone so there is so much that needs to be done to make people aware, so young females know nothing about PrEP which is the reason they don't trust it or even request it" [FG:3, P22, Male, Age: 45]*

### Health promotion barrier: Inaccurate information dissemination regarding PrEP

Registered nurses further alluded that one of the barriers contributing to young people not coming to take PrEP is inaccurate information dissemination regarding PrEP as they shared that:

*"What I have noted is that, there is an incorrect information out there regarding PrEP that we do not know, you know people always create stories, I had 3 to 4 patients who came back with PrEP saying, people say it makes you lose weight because it eats your liver...[laughs]…" [FG: 2, P19, Female, Age: 35]*

Inaccurate information dissemination regarding PrEP strongly came out from several participants as they encountered patients rejecting or declining the initiation of PrEP and or returning the pill to the clinic stating that there is health implication information circulating regarding PrEP which instil fear in them. Some young females believe that PrEP it actually gives you HIV rather than preventing it:

*"Many of honest patients bring them back shame, saying this things people says it actually gives you HIV…"* [FG: 4, P20, Female, Age: 39]

### Health promotion barrier: HIV-ART related stigma

HIV-ART related stigma is still a public health concern which is the reason often patient decide to use the clinic far away from their homes because no one will know them and take either ART or PrEP there. As such, young females are uncomfortable to freely take PrEP as they will be labelled HIV positive. Participants 18 mentioned that:

*"The problem is, whether we like it or not, HIV is still stigmatized and so is PrEP as they are closely related, for example, in this clinic we help people from far, why? because they are scared to go to the clinics near their home because once people see you taking PrEP, they think you are HIV positive already and people opt not request it."* [FG: 3, P18 Female, Age: 36]

This implies that, PrEP usage is directly affected by HIV or ART related stigma and the fact that PrEP pill is made up of Tenofovir disoproxil fumarate and emtricitabine which are drugs used in some combination of ART for HIV positive patients. Instill the fear of being labelled HIV positive. Participants mentioned that:

*"To many patients says PrEP is the same as ART and this is because PrEP is made up of ARTs, so they search and therefore afraid to request it with fear of being labelled HIV positive"* [FG: 5, P 25, Female, Age: 37]

### Medication related barriers

This sub-theme emerged from focus group discussions for registered nurses who alluded that, there are PrEP medication related barriers contributing significantly to low number of young females either accepting or being retained in PrEP pill for as long as they are still at risk of HIV. This sub-theme emerged with three categories which included: PrEP pill side effects, contra-indications for PrEP initiation, PrEP follow-up and monitoring burden. These categories are outlined in detail below:

### Medication related barrier: PrEP pill side effects

PrEP pill side effects emerged as a pill related barrier which often hinders young females to request PrEP or even continuously take it. Mainly because when they feel sick following PrEP initiation, they stop it by themselves and others would not even accept it when they hear that it has side effects. Participants shared that by saying:

*"They often talk about the headaches and gaining weight or nausea, these makes them to stop taking PrEP and they talk, so they tell others who wanted to start it then no one will come for PrEP"* [FG: 3, P 19, Female, Age: 38]

Comparably, the other participants stated that young females stop PrEP immediately when they experience PrEP side effects:

*"Side effects is our main problem, once they hear about it or even experience it, if already on PrEP, they stop it"* [FG: 5, P 30, Female, Age: 44]

### Medication related barrier: Contra-indications for PrEP initiation

In this category, registered nurses perceived that there is PrEP related barrier of contra-indications to prescribe PrEP pill, stating that PrEP pill is only made up of two ART drugs with no other option for patients with other chronic conditions such as kidney dysfunction:

*"Let's talk about young females who does not qualify to be on PrEP because of Tenofovir [TDF], isn't that when your kidney functioning is not good you can't take TDF and remember PrEP has TDF, and there is no alternative, it is just standard, so it means in such case I can't give PrEP"* [FG: 2, P 12 Female, Age: 31]

This suggests that, limited options on combination of PrEP pill hinders registered nurses to offer PrEP to a wide range of patients with different chronic conditions.

### Medication related barrier: PrEP follow-up and monitoring burden

In regard to the follow-up and monitoring burden of PrEP services, it was indicated that monthly or two monthly PrEP pill follow up visit to the clinic by patients increases nurses daily workload. As a results, nurses are often reluctant to promote PrEP services. Increased workload standing out as a main concern of nurses as indicated:

*"The fact that these patients who are initiated on PrEP must come every month or every second month, is just too much guys…so thinking about that, we get reluctant to promote it because we will be creating more work load for ourselves"* [FG: 2, P 14, Female, Age: 39]

Into the bargain, monitoring of patients on PrEP medication is continuing to be a burden to nurses in different healthcare facilities as stated that, HIV testing should be performed every visit of patients on PrEP additional to medication collection. The other participants mentioned that:

*"We supposed to test HIV for patients on PrEP all the time, I mean why young people have to come here every month and worse part, not only collecting medication but proper assessment and HIV testing should be done, that is just too much for us nurses"* [FG: 2, P 10, Female, Age: 55]

### Theme 2: Strategies to enhance utilization of PrEP amongst young females

In this theme, strategies to enhance utilization of PrEP amongst young females were explored as perceived by registered nurses. These strategies were categorized into the following sub-theme; healthcare structure, healthcare provider, health promotion and PrEP medication related strategies. Each of these sub-themes had categories presented in detail below. The verbatim transcribed focus group discussion for registered nurses has been quoted under each strategies to enhance utilization of PrEP amongst young females as outlined in the S1 File. The data was collected using the S2 File. Focus group discussion guide for professional nurses during focus group discussions held.

### Healthcare system structure strategies

This sub-theme emerged from the findings of this paper suggesting that there are healthcare system structural strategies to enhance the use of PrEP both the utilization. This sub-theme had four categories which included increased PrEP accessibility, integrate PrEP into HIV services, integrate PrEP into family planning services and partnership with Department of Education. These categories are outlined in detail below:

### Healthcare system structure strategy: Increased PrEP accessibility

Participants perceived that there are healthcare system strategies which could contribute to improved PrEP utilization. The mentioned strategies included increased PrEP accessibility in every or nearby clinics rather than few clinics offering PrEP as verbalised by Participant 8:

> *"Government must bring the medication closer to where people are, I mean it is not in every clinic, so young people must use maybe four taxis to get prevention medication, people want services near them…" [FG: 2, P 8, Female, Age: 34]*

Implicit in the above statement is that the distance to the clinic that is fully capacitated to offer PrEP should be reduced by making PrEP available in all the clinics. Suggesting that, address all the challenges that causes other clinics not to offer PrEP and avoid referring patients from one clinic to another. Another participant concurred and stated that:

> *"…People are tired to attend far clinics and even us, we are tired to help patients for other clinics, government must make everyone aware that they can get PrEP everywhere if already those clinics are implementing it and if not, they must implement it…" [FG: 3, P 18, Female, Age: 51]*

### Healthcare system structure strategy: Integrate PrEP into HIV services

Integrating PrEP into HIV services was perceived as one of the quick winning strategies mainly because in over 90% of clinics, HIV services are well established and therefore, it would be beneficial to integrate PrEP in such services to receive maximum usage. One of the participants mentioned that:

> *"… we are having flyers for HIV testing, what if the department include PrEP on the same flyer so that people can link the two and believe that this is true. Healthcare workers going to the township to do HIV testing should give people those flyers when they test HIV to achieve two objectives in one community outreach which will include PrEP and HIV services" [FG: 1, P 7, Female, Age: 51]*

Additional to that participant 39 also suggested that

> *"All the HIV testers must be given target on how many patients they referred for PrEP initiation from those whom they tested HIV negative, then you will see numbers of those coming to be initiated on PrEP will increase significantly." [FG: 3, P 17, Female, Age: 43]*

### Healthcare system structure strategy: Integrate PrEP into family planning services

PrEP must be integrated into family planning to increase the number of young females accepting to be initiated on PrEP mainly because they will be coming to prevent unplanned

pregnancy suggesting that they are sexually active and need PrEP to prevent acquisition of HIV. Other participants stated that:

*"PrEP and FP must be given in the same room because that's where sexually active girls are…" [FG: 4, P 23, Female, Age: 39]*

While the other participants stated that, PrEP should be offered in one consulting room with FP mainly because the clinician in that room is often well experienced to work with young people and will be able to educate them in a way that young females will see the need of PrEP. Additional that, to reduce the number of queues young women follow in the clinic which also increase patients waiting time, integration of PrEP and FP would be one of the best strategies:

*"Having so many departments in the clinic will not help us, I mean we are having family planning room already where these girls go to all the time and someone who knows how to work with these patients is there already, just give it to those nurses, you will see. PrEP will improve, patient will not agree to go to many consulting rooms rather let's combine these rooms, PrEP, and FP" [FG: 3, P 19, Female, Age: 42]*

### Healthcare system structure strategy: Partnership with Department of Education

Department of health's partnership with department of education may improve PrEP utilization amongst young females. Participants mentioned that *PrEP campaigns should be conducted at schools where young women are*:

*"Universities and schools must play their role as well, or let me say, it should be made clear that they have a role to play in prevention of HIV. I have never heard of any PrEP campaigning at schools, yes, I think that will help" [FG: 5, P 30: Female, Age: 49]*

Furthermore, participants stated that PrEP should be made part of department of education syllabus, to increase awareness and make young females aware that, should they require detailed information on PrEP, they are advised to visit the clinic:

*"I think also schools can really assist by allowing us to do PrEP campaigns there. Or even teach kids about PrEP even if it can be a little bit of information, just to make them aware that there is this thing called PrEP. Included PrEP into their syllabus as much as they teach about HIV" [FG: 2, P 9, Female, Age: 30]*

### Healthcare provider related strategies

Participants alluded that, there are healthcare provider related strategies which could contribute to increased PrEP utilization rate amongst young females. This sub-theme emerged with two categories which included: proactive management of side effects and capacitate nurses on PrEP guidelines. These categories are outlined in detail below:

### Healthcare provider related strategy: Proactive management of side effects

Nurses should proactively manage side effects so that young females do not stop taking PrEP pill after being initiated on it:

*"Like what we do with pyridoxin, lets prevent side effects before they occur, give them PrEP and those preventative medication for side effects." [FG: 1, P 1, Female, Age: 38]*

Participants emphasized that patients should be informed about the side effects so that they know what to expect before they experience them. Mainly because it be a shock if they were not expecting them (side effects) and that will assist in preparing young females mentally. Participant 33 also shared that:

> "*Let's give them remedies on how to overcome side effects, or even medication like Paracetamol. When they leave the clinic with PrEP, they must also leave with some prevention of side effects medication*" [FG: 4, P 33, Female, Age: 39]

While other participants mentioned that nurses should be transparent on how young females might feel after the first or second dose of PrEP and how to treat those effects at home.

**Healthcare provider related strategy: Capacitate youth clinic nurses on PrEP guidelines.** Nurses must be capacitated on PrEP to enhance the quality of HIV PrEP services which in turn will improve utilization of young females on PrEP:

> "*…there should be training programs for us nurses, they must teach us, you can't have 4 people trained on PrEP out of 25 nurses, train us, mentor us, support, take us to refresher trainings. Just train nurses on PrEP*" [FG: 2, P 15, Female, Age: 43]

It was recommended that every clinic have a specific PrEP small clinic with well capacitated nurse on PrEP because young females often get frustrated when they must queue with all other patients in the clinic. To ensure that PrEP clinics are functioning sufficiently, nurses should be capacitated on PrEP as a program and the other participant mentioned that:

> "*We as nurse it's a shame that some of us were never trained on PrEP, we need training, training must be for all nurses so that when patients ask me, I can answer than to always say go that side they will explain, it makes us feel less before patients*" [FG: 2, P 14, Female, Age: 44]

## Health promotion related strategies

Participants indicated that, there are health promotion strategies which would contribute significantly to the improved PrEP utilization rate. This sub-theme emerged with three categories which included: PrEP awareness, sensitization through media, Information dissemination through community outreach program, high-quality counselling, and intensified PrEP health education. These categories are outlined in detail below:

## Health promotion related strategy: PrEP awareness and sensitization through media

To increase awareness of PrEP as the best HIV prevention intervention, media should be used to spread the information and sensitize general population including young females on PrEP:

> "*Most of young people, girls and boys are on social media, they love TV, so make people aware of PrEP using those platforms*" [FG: 2, P 15, Female, Age: 47]

The other participants mentioned that media platforms can be used to increase awareness since both young females and general population have access to one or two of the platforms:

> "*On TV as well, we can have dramas that educate both young people and their parents about PrEP because, you know, everyone has a television these days, radio, newspaper, SABC one stories, we have stories and many people watch them, put it there to educate more people…*" [FG: 2, P 12, Female, Age: 41]

### Health promotion related strategy: Information dissemination through community outreach program

Registered nurse indicated that community outreach programs ought to be implemented to reach out to everyone, who is not on social media or does not have radio or television, with PrEP information:

*"I think if government can expand the objectives of Ward-Based Outreach Team, as they do door to door, they must have a nurse giving information about PrEP and initiate on spot if possible"* [FG: 2, P 9, Female, Age: 32]

While others shared that PrEP campaigns can be done at malls where they would erect a gazebo with some music to attract young females for recruitment of PrEP easily:

*"Let's go to where people are and tell them about PrEP, community door to door or recreational centers such as football playgrounds or any sports activities or malls"* [FG: 5, P 28 Female, Age: 38]

### Health promotion related strategy: High-quality counselling and intensified PrEP health education

Registered nurses stated that amongst other strategies to increase the number of young females requesting PrEP and remaining in it, is high-quality counselling and intensified health education on PrEP. To address lack of information on PrEP, quality counselling came out strongly from several participants:

*"Let's put ourselves in patients' shoes, there is no way I cannot take PrEP when I'm not told that this will prevent me being infected with HIV in detail, the problem is how we are telling them, is either nobody says anything to them, or we just say it in passing. If they ask questions, we eat them alive [laughs]…so let's give in-depth and quality counselling or PrEP education"* [FG: 1, P 6, Female, Age: 48]

Participant 28 also shared that

*"I agree, proper education and counselling will help to make them aware of PrEP and be able to zoom in their personal challenges to take PrEP and help where possible or involve different stakeholders to intervene should there be a need"* [FG: 1, P 2, Female, Age: 46]

### Medication related strategies

This sub-theme emerged from focus group discussions where participants alluded that, there are PrEP medication related strategies to increase number of young females either accepting or being retained in PrEP pill for as long as they are still at risk of HIV. This sub-theme emerged with two categories which included: convenient mode of administration and additional PrEP pill combination. These categories are outlined in detail below:

### Medication related strategy: Three monthly injectable PrEP

Participants indicated that convenient mode of administration for PrEP may increase the number of young females going to the clinic to request PrEP or even remaining on it if already taking it. This is mainly because it has been proven previously that young females are

compliant to three monthly injectable for FP and therefore would be ideal for them to have the same convenient mode of administration:

*"None of us can be compliant in simple things like seven days antibiotics, how do we expect the poor kids to comply especially because they are not sick, so PrEP injection every three months will be the best"* [FG: 2, P 11, Female, Age: 39]

Another participant indicated that, with PrEP injectables young females will not be required to visit the clinic every month and will not take a pill everyday:

*"These kids hate taking tablet every day, so let's give them PrEP in a form of injection just like FP and they must return in two or three months"* [FG: 5, P 30, Female, Age: 51]

### Medication related strategy: Additional PrEP pill combination

Participants further indicated that, additional PrEP pills combination may be another strategy to increase PrEP utilization mainly because of limited option they currently have which does not accommodate for wide range of patients such as patients with kidney dysfunction:

*"We have only one PrEP pill made of Tenofovir and emtricitabine, why are we not having alternatives PrEP pill forms like in the ART program. So that if you have kidney problem and you can't take Tenofovir we then give you second option"* [FG: 3, P 22, Female, Age: 38]

Suggesting that, should the young females be diagnosed with kidney functioning, they would not be able to take PrEP and therefore having another form of PrEP pill combination which will accommodate wide range of patients is crucial.

## Discussion

Many young women in Gauteng Province face barriers of limited access to PrEP healthcare services including distance, cost, and lack of transportation. This can make it difficult for them to receive PrEP and to adhere to the recommended dosing schedule. Implying that, in this setting access to PrEP affects PrEP usage significantly as one of the healthcare system structural barriers indicating that often young females are required to travel longer distance to access PrEP in a specific clinic. These findings are consistent with previous studies on the distance to the clinic being a significant predictor to HIV PrEP uptake and follow-up visits to the clinic [22]. Suggesting that, adequate PrEP healthcare services access is compromised. Additional to that, participants stated that young females often visit one of the clinics to request PrEP and would either be referred to the other clinic mainly because unavailability or limited of PrEP trained nurse thus patients would wait for long or rather leave the clinic without PrEP pill. Implying that ineffective workflow and shortage of nurses serves as barrier to good PrEP utilization.

Comparably, insufficient total number of nurses available to offer PrEP continues to be a crucial challenge in PrEP implementation processes mainly because PrEP initiation and or follow-up care would be impossible [26]. As such, low number of nurses trained on PrEP and their competency thereof is one of major concerns around PrEP wherein participants indicated that, out of fifty plus registered nurses, only four [3] were trained on PrEP (8%) while in other clinics about 80% of nurses were not trained on PrEP. These findings confirm those of [6] whose study reported that training of healthcare providers, including nurses, has been

reported to be concerning in the roll out of PrEP and this affects HIV PrEP utilization across key population groups [27]. On that bargain, the ethical dilemmas that registered nurses often find themselves in whereby encouraging young females to take PrEP is almost impossible mainly because they regard "15-year-old young females" as a child or even granddaughters and it is as if they are giving them a permission to have multiple sexual partners indicating that it morally unacceptable and compromises their values as human beings. Consistently [21] found that many patients still connect or associate HIV/AIDS-related programs with moral decadence and or moral judgement of those directly or indirectly involved. That has been found to be one of the factors affecting the retention of a patient in care [28].

PrEP is fairly new in many settings, however the awareness strategies put in place are not adequate to cover a wider population hence young females do not come to the clinic to request it. [22] concurred when their study reported a very low PrEP awareness among Adolescent girls and young women who stated that they heard about PrEP for the first time at the clinic during the time of their study [24,25]. The poor awareness is evident on an inaccurate information or reports disseminated by patients regarding PrEP. These reports indicate uncertainties and inaccurate PrEP information dissemination which contribute to low HIV PrEP utilization amongst young females in the Gauteng province. These submissions resonate with [29] whose study reported that participants referred to the PrEP pill as "a problematic due to severe side effects and demanding pill". Similarly, [30] found that PrEP users discontinued taking PrEP because of the side effects they experienced. This implies that the barriers explored from registered nurses at this specific setting concurs with findings of authors across the globe.

Over and above that, contra-indications for PrEP initiation also affect high HIV PrEP utilization mainly because PrEP pill is a combination of only two drugs, Tenofovir (TDF) and emtricitabine (FTC) with no alternatives. Implying that when young females do not qualify to be on Tenofovir (TDF) because they have kidney dysfunction, they cannot be given PrEP or rather sensitive emtricitabine. Compatible with [27] who reported that lack of alternatives to PrEP can be a hindrance to high HIV PrEP utilization [31]. While PrEP follow-up and monitoring burden is a concern mainly because patients initiated on PrEP are required to visit the clinic every month for their follow-up care which often increases nurses' workload. Hence nurses are often reluctant to recruit more patients including young females to take PrEP. This submission resonates with [5] who indicated that a high number of patients who defaulted PrEP were influenced by inadequate clinical resources for routine screening and monitoring as patients would be returned home and requested to return in two or three days. Not to mention the HIV-ART related stigma which was expressed as the "giant" affecting PrEP utilization. Mainly because, often general population confuses PrEP pill with ARTs which create more deterrent to PrEP use because of the fear of being ostracized by their communities. The confusion between ARTs and PrEP pills is closely related to similarities noted such as packaging, size and demand to take it daily at night [24,27,30,32,33].

To overcome these barriers, there are several strategies that can be employed by healthcare providers and policymakers to improve the utilization of PrEP among young females in Gauteng Province. These strategies include increased PrEP accessibility which is a critical healthcare system structural strategy to ensure that young females received PrEP services at any given time in all the public healthcare institutes. This is to reduce the burden on either nurses or clinics to offer PrEP services to young females who are not within their clinic demarcation. Implying that, interventions to increase the accessibility of PrEP should be in place to improve HIV PrEP utilization [22,27,30,34,35]. Additional to that, training of nurses on PrEP guidelines may be one of the solutions to increase number of nurses who can offer PrEP, thus improving access. Consistent results were reported by File32] whose study

indicated that PrEP training for healthcare providers is required to scale-up the provision of PrEP to the key populations including young females [36].

Over and above that, integration of PrEP into HIV services could improve PrEP utilization wherein the current HIV testing flyers includes PrEP information and how it works so that patients can be knowledgeable on the two [HIV testing and PrEP] concurrently to further improve the authenticity of their relation. Comparably, [34] indicate that incorporating oral PrEP pills into a well-established HIV service for South African women will be a convenient strategy for PrEP performance improvement [37]. Additional to that, family planning services was noted as another stream of integration for PrEP as a strategy to improve performance mainly because those "young females" visiting the clinic for FP could be sexually active and therefore at increased chance of HIV acquisition. Consistent results were reported by various studies which indicated that incorporating PrEP pill services into well-established standard prevention services for women is essential for good HIV PrEP utilization [3,26,24,37,38]. Apart from that, [37] assessed facilitators to achieve a "win-win" performance for both FP and PrEP services and discovered that PrEP-FP integration has the probability to achieve both PrEP and FP good performance [38]. On that bargain, Department of Education syllabus should include PrEP education in detail since they currently teach about HIV. Therefore, PrEP as part of HIV-related programs, should be included to inform young people about the availability, indication and importance of the PrEP pill. Similar findings were reported by [29] whose study indicated that the involvement of stakeholders such as the Department of Education is a key to reaching out to as many young people as possible.

Proactive management of side effects is critical as most young females discontinue PrEP pill is due to the side effects therefore, addressing PrEP pill side effects proactively would either prevent them or manage without stopping medication [24,27,39]. Moreover, high-quality counselling and intensified PrEP health education are significant tools towards PrEP awareness and addressing in-depth personal challenges of young females to take PrEP. The same results were reported by several studies that indicated that thorough high-quality counselling and intensified PrEP health education, all challenges young people face when they want to take PrEP can be addressed. [2,24,32,40,41]. Additional to that, PrEP awareness and sensitization through media wherein television or radio advertisements are recommended as helpful in legitimizing explanations of PrEP to the population at large including parents of young females because they often do not have access to social media [27,32,40,42,43]. In addition, [5] reported that awareness and PrEP knowledge are masterpieces of successful HIV PrEP uptake and continuous follow-up amongst key populations. Furthermore, information dissemination through community outreach program by incorporating the PrEP program into standard and well-stablished community outreach programs is important to attaining high HIV PrEP utilization amongst key populations such as young females who are often not visiting public health institute for PrEP services [23,29,32,37].

One of critical measure to curb all the concerns around taking the PrEP pill daily is a quarterly injectable mode of PrEP administration which would be intramuscular eliminating concerns regarding daily taking of a pill and monthly clinic's visit and thus reducing waiting time in different clinics. These submissions confirm what [22] and [44] findings, whose studies indicated that injectable PrEP is the ideal mode of administration for many key population groups including young females which will be convenient and also addresses stigma related concerns. Over and beyond what has been discussed, only one PrEP pill combination made of Tenofovir (TDF) and emtricitabine [FTC] availability is a hindrance to offer PrEP services and therefore, alternative PrEP pill combination is required to accommodate a wide range of patients replicating the ART drugs options. Similarly, [23] indicated that different kinds of patients diagnosed with different non-communicable diseases should be accommodated on

PrEP options in case of contraindication such as hypersensitivity, drug to drug interaction and or the condition that the patients suffer from [45].

## Conclusion

The healthcare system plays a crucial role in providing preventive care and treatment to individuals, including young females. However, there are several barriers that hinder the utilization of PrEP among this population in Gauteng Province from registered nurse's perspective. Addressing barriers to access, healthcare worker related such as training, competency and staffing is essential to improving the utilization of PrEP. Over and above that, health promotion barriers which included poor awareness, inaccurate PrEP information, HIV-ART related stigma and PrEP pill related such as side effects, contraindication and monitoring burden are crucial to address enhance PrEP usage. Moreover, implementing strategies such as increased access, PrEP integration to existing HIV, FP services, Department of Education, and training of nurses on PrEP would assist to overcome these barriers. By doing so, registered nurses would contribute significantly to reduce the incidences of HIV/AIDS among young women in Gauteng Province, thus improving the overall public health outcome in this region. Further studies could be done to explore perceptions of young females regarding PrEP utilization. To understand their primary challenges, taking it into consideration that their "voice" is golden to address all the PrEP utilization challenges from the user's perspective.

## Supporting information

**S1 File.** This is SI Fig transcribed and analysed focus group discussion data for registered nurses.
(PDF)

**S2 File.** Focus group discussion guide for professional nurses.
(DOCX)

## Acknowledgement

We would like to thank all the clinics that took part in this study and the co-coder - Professor A van der Wath who confirmed the themes, sub-themes and categories emerged in this study after analysing focus group discussion data from registered nurses in Gauteng province.

## Author contributions

**Writing – original draft:** Doreen Onkarabile Mudau.

**Writing – review & editing:** Doreen Onkarabile Mudau, Fhumulani Mavis Mulaudzi, Nombulelo Veronica Sepeng, Rafiat Anokwuru.

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
