## [Decision Letter · Decision Letter 0]

12 Jul 2023

Dear Dr. Mudau,

Thank you for submitting your manuscript to PLOS ONE. After careful consideration, we feel that it has merit but does not fully meet PLOS ONE’s publication criteria as it currently stands. Therefore, we invite you to submit a revised version of the manuscript that addresses the points raised during the review process.

Address comments on the methodology as raised by reviewer #1.

We look forward to receiving your revised manuscript.

Kind regards,

Sogo France Matlala, PhD

Academic Editor

PLOS ONE

Journal Requirements:

Reviewers' comments:

Reviewer's Responses to Questions

**Comments to the Author**

1. Is the manuscript technically sound, and do the data support the conclusions?

Reviewer #1: Yes

Reviewer #2: Yes

2. Has the statistical analysis been performed appropriately and rigorously?

Reviewer #1: N/A

Reviewer #2: Yes

3. Have the authors made all data underlying the findings in their manuscript fully available?

Reviewer #1: No

Reviewer #2: Yes

4. Is the manuscript presented in an intelligible fashion and written in standard English?

Reviewer #1: No

Reviewer #2: Yes

Reviewer #1: Reviewer’s comments

The manuscript is technically sound, and having the data that support the conclusions.

The manuscript is qualitative in nature.

The abstract:

The conclusion should state the main findings first and then talk about implications and the recommendations.

Introduction

The introduction needs to be restructured according to journal requirements. The problem is not clearly outlined. The researcher should elaborate more on what is known or unknown about the subject (Healthcare Systems Barriers and Strategies for Pre-Exposure Prophylaxis Utilization amongst Young Females: Registered Nurse’s Perspective) and relate it to other studies conducted globally, especially in the African continent.

Methods

The methodology used was appropriate for the study and well explained. But there was no methodological orientation stated to underpin the study

What type of method was used to approach the study participants?

How many people refused to participate or dropped out of the study?

How was the study setting? it needs to be clearly described.

What type of data collection instrument that was used in this study? was pre-testing or pilot test ever conducted in this study?

Were repeat interviews carried out in this study? If yes, how many?

It was not mentioned if the researcher has used audio or visual recording to collect the data, or if Field notes were made during and/or after the focus group?

The issue of data saturation was never mentioned in this study.

How many data coders have coded the data?

The whole of methodology needs to be re-arranged so that it can flow accordingly. Recruitment of the participants cannot happen after participants have signed the consent forms.check line number 135-142.

Reporting:

Each quotation needs to be identified .e.g. FG number,participant number,Sex and Age.

Some of the quotations were not provided - e.g line number 255:just to say 15 registered nurses, is not sufficient enough without providing some quotations.

The authors should take note of the typos, spelling, grammar, and phrasing issues so that the document can be readable.

Conclusion

The conclusion should highlight the main findings you want to communicate, not a summary of the result. Then make a few recommendations.

Reviewer #2: The manuscript is technically sound, written well and the data support the conclusions. However, minor corrections listed below are to be made;

1. Under materials and methods

Lines 113-114: Correct 28h to 28th, then 1ST to 1st.

Line 135: The ethical approval as well as other approvals should be the first statements under materials and methods.

Line 142: Which of the 5 clinics were considered in the study?

Results

Lines 150-155: use either n or N but consistency must be maintained.

Discussion

Authors may also discuss the implications of HIV drug resistance to the utilisation of PrEP to those who wont be retained in care and are at risk of infection.

**Do you want your identity to be public for this peer review?** For information about this choice, including consent withdrawal, please see our Privacy Policy

Reviewer #1: No

Reviewer #2: No

---

## [Author Response · Author response to Decision Letter 1]

26 Aug 2023

Thank you so much for the valuable comments to make this manuscript more meaningful, we are looking forward to positive feedback. Thank you

---

## [Decision Letter · Decision Letter 1]

27 Oct 2023

Healthcare Systems Barriers and Strategies for Pre-Exposure Prophylaxis Utilization Amongst Young Females in Gauteng Province: Registered Nurse’s Perspectives

PONE-D-23-09915R1

Dear Dr. Mudau,

We’re pleased to inform you that your manuscript has been judged scientifically suitable for publication and will be formally accepted for publication once it meets all outstanding technical requirements.

Kind regards,

Miquel Vall-llosera Camps

Senior Staff Editor

PLOS ONE

Reviewers' comments:

Reviewer's Responses to Questions

**Comments to the Author**

Reviewer #1: All comments have been addressed

Reviewer #2: All comments have been addressed

2. Is the manuscript technically sound, and do the data support the conclusions?

Reviewer #1: Yes

Reviewer #2: Yes

3. Has the statistical analysis been performed appropriately and rigorously?

Reviewer #1: N/A

Reviewer #2: N/A

4. Have the authors made all data underlying the findings in their manuscript fully available?

Reviewer #1: Yes

Reviewer #2: Yes

5. Is the manuscript presented in an intelligible fashion and written in standard English?

Reviewer #1: Yes

Reviewer #2: Yes

Reviewer #1: Because the authors sufficiently addressed the issues brought up during a previous round of review, I think the manuscript is now appropriate for publishing.

-The article presents a scientific investigation that is technically sound and includes supporting data. The right conclusions were drawn using the information that was presented.

-The data should either be incorporated into the publication or stored in a publicly accessible source.

-.All of the evidence that supports the conclusions in the authors' publication has been fully disclosed.

-The language used to present the material is precise, direct, and clear.

Reviewer #2: The authors were able to address all comments to satisfaction, the manuscript is recommended for publication.

**Do you want your identity to be public for this peer review?** For information about this choice, including consent withdrawal, please see our Privacy Policy

Reviewer #1: No

Reviewer #2: No

---

## [Editor Report · Acceptance letter]

PONE-D-23-09915R1

PLOS ONE

Dear Dr. Mudau,

I'm pleased to inform you that your manuscript has been deemed suitable for publication in PLOS ONE. Congratulations! Your manuscript is now being handed over to our production team.

Kind regards,

on behalf of

Dr. Miquel Vall-llosera Camps

Staff Editor

PLOS ONE